# Building capacity of primary health care workers and clients on COVID-19: Results from a web-based training

Olumuyiwa O. Odusanya[1]*, Adeyinka Adeniran[1], Omowunmi Q. Bakare[2], Babatunde A. Odugbemi[1], Oluwatoyin A. Enikuomehin[2], Olugbenja O. Jeje[3], Angela C. Emechebe[3]

1 Department of Community Health and Primary Health Care, Faculty of Clinical Sciences, Lagos State University College of Medicine, Ikeja, Lagos, Nigeria, 2 Department of Computer Science, Faculty of Science, Lagos State University, Ojo, Lagos, Nigeria, 3 Department of Community Health and Primary Health Care, Lagos State University College Teaching Hospital, Ikeja, Lagos, Nigeria

* olumuyiwa.odusanya@lasucom.edu.ng

## Abstract

### Background

Health care workers (HCWs) in the first line of care play critical roles in providing the correct information about the coronavirus disease to the community. The objective of the study was to determine the effect of virtual training on the knowledge, attitude, and preventive practices among PHC workers and their clients in the prevention and control of coronavirus disease.

### Methods

A quasi-experimental intervention virtual training, using a before and after design amongst HCWs and clients was conducted at primary health care facilities in two Local Government Areas of Lagos State. The study instruments were pre-tested questionnaires for both HCWs and their clients. which investigated knowledge of symptoms, modes of disease transmission, methods of prevention, and preventive practices. Changes in knowledge, attitudes, and practices were compared pre-and post-intervention. The level of significance was set at $p < 0.05$.

### Results

Sixty-three HCWs (out of 100 recruited at baseline) and 133 clients (out of the initial 226) completed the study. The mean ages of the HCWs and clients were 39.2±9.9 and 30.9±5.0 years respectively. At the baseline, the HCW's knowledge was good in the domains of symptoms, modes of transmission, and preventive measures. The training led to a higher but not significant ($p> 0.05$) increase in the level of knowledge. Contact with trained HCWs was found to lead to significantly ($P < 0.001$) higher levels of knowledge, attitudes, and preventive practices. amongst clients.

**Data Availability Statement:** All relevant data are within the article and its Supporting Information files.

**Funding:** OOO received a research grant from Lagos State Research and Innovation Council. website. https://lasric.lagosstate.gov.ng The funders had no role in study design, data collection and analysis, decision to publish, or presentation of the manuscript.

**Competing interests:** The authors have declared that no competing interests exist.

## Conclusion

The training was effective in improving the knowledge of both the trained HCWs and their clients.

## Introduction

The current pandemic caused by novel Coronavirus (2019-nCoV), known as coronavirus disease (COVID-19) was first reported in December 2019, as a cluster of acute respiratory illnesses in Wuhan, Hubei Province, China, from where it spread rapidly to all parts of the world [1]. As of August 26, 2022, the number of confirmed cases reported globally to the World Health Organization was 596,873, 121 confirmed cases, and 6,459,684 deaths [2] and in Nigeria, the number of confirmed cases is 263,090 with 3,148 deaths [2]. Effecting behaviour change about COVID-19 is partly dependent upon the level of knowledge possessed by health care workers (HCWs) and the community. Various studies have been conducted to determine the level of knowledge in both groups.

In the United Arab Emirates (UAE), a study amongst 453 HCWs showed that 61% had a poor level of knowledge of the means of transmission, 64% had poor knowledge of symptoms and 75% had positive perceptions about the disease. Age and professional cadre were associated with inadequate knowledge and poor perception [3]. In Yemen, a study amongst 514 HCWs found that physicians had statistically significantly higher levels of knowledge than other professional groups (p < 0.001). HCWs in intensive care units and emergency departments were better prepared than in other areas of care probably because of continuing exposure and training about the disease [4].

A group of workers from Mozambique reported knowledge of COVID-19 amongst HCWs. They found that a quarter correctly identified all three common symptoms (shortness of breath, dry persistent cough, and fever), 2% correctly identified groups at risk, 16% knew three means of transmission and 39% knew three methods of prevention. The response rate in the study was 28% [5]. In Uganda, 69% of HCWs had sufficient knowledge, and 74% reported practising good preventive measures [6]. In Ethiopia, a facility-based cross-sectional study amongst 379 HCWs using multi-stage sampling techniques reported that 75% had adequate knowledge, 84% had positive attitudes and 69% had good preventive practices. Factors associated with good knowledge were higher levels of education (a master's degree), training on COVID-19, and working in a comprehensive specialist hospital. Good preventive practices in the study were associated with the source of information, working in primary hospitals, and possessing good knowledge. Work experience and knowledge were significantly associated with attitudes [7].

Nine out of ten HCWs in a teaching hospital in Bayelsa State, Southern Nigeria had good knowledge, 90% had good hand hygiene practices but 52% used face masks [8]. This is not unexpected due to the concentration of highly skilled and knowledgeable HCWs in teaching hospitals. The proportion of HCWs with good knowledge in a study from Northern Nigeria was 83%. Eight out of ten had positive attitudes and 88% had good preventive practices. The odds of good knowledge were significantly lower among community health officers compared with doctors. Positive attitudes were predicted by good knowledge, older age (fifth decade), higher education, and Christian faith. A significant association was also found between good preventive practices and good knowledge [9].

A study amongst 589 participants recruited using snowball sampling in North-Central Nigeria reported that 99.5% had good knowledge of COVID-19. The source of information

was through social media (56%) and television (28%). Almost 80% had positive attitudes and 82% used face masks consistently. A significant relationship was found between knowledge of COVID-19 and attitudes towards preventive practices. A limitation of the study was that only literate subjects and those with internet access could participate in addition to the use of non-probability sampling methods [10].

A study amongst 961 young Nigerians showed that 92% had good knowledge about clinical symptoms of COVID-19 such as fever, fatigue, dry cough, and body pains but only 31% reported wearing face masks. A strong association was found between knowledge and age groups. A higher level of education was found to be significantly associated with the use of preventive measures such as wearing face masks and regular hand washing [11]. In Kano, Northern Nigeria, 30% of respondents had good knowledge, 11% had good attitudes and 26% had good preventive practices, which are lower than values in other parts of Nigeria [12].

A cross-sectional study amongst mothers of under-five children attending a teaching hospital in Enugu showed that 65% were aware that asymptomatic persons could transmit COVID-19, 72% thought children under five years could contract the infection and 43% had good perceptions about COVID-19 [13]. Eighty-six percent of pregnant women (n = 527) in Northern Ghana were found to have adequate knowledge about COVID-19 and 49% had positive preventive practices. Factors associated with adequate knowledge were having at least a primary six education, living in an urban area, and having received health education about the disease at a health facility. Having good preventive practices was associated with having at least a primary six education, living in an urban area, and pregnant women with a chronic disease while multiparity was negatively associated with it [14].

With varying levels and gaps in the knowledge and practices of HCWs and the public, it becomes important to build the capacity of both groups if COVD-19 is to be effectively controlled. HCWs at the first level of care attend to the larger segments of the population and are in the best position to influence them. Positioning HCWs to be able to assume this role requires continuous training. Formal means of training such as face-to-face contact lectures may not be suitable during the pandemic. Thus, a virtual training approach becomes useful. Moreover, there has been very limited training for HCWs, especially at the first level of care [15]. Therefore, the goal of the study was to determine the knowledge, attitudes, and preventive practices of HCWs, train them and assess the effect of the training on their clients' knowledge, attitudes, and preventive practices towards the infection.

## Materials and methods

### Settings

Lagos State is one of the 36 states in Nigeria and remains the economic nerve centre of the nation. Politically, it has 20 Local Government Areas (LGAs), 37 Local Council Development Areas (LCDAs) and 376 Wards. The population is estimated to be over 20 million [16]. Health services for the citizens are available through both public and private facilities. The larger numbers of publicly owned facilities in Lagos State belong to the Lagos State Government although there are a few owned by the Federal Government. The Primary Health Care (PHC) level of care is managed by the Lagos State Primary Health Care Board.

The study was conducted in Ikeja and Alimosho local government areas (LGAs) of Lagos State. Ikeja LGA is the capital of Lagos State. The LGA is bound in the north by Ifako-Ijaiye LGA, the south by Oshodi-Isolo LGA, the east by Agege LGA, and the west by Kosofe LGA. The LGA has an estimated population of 460,000 residents. There is a mixture of public and private health facilities in the LGA. Government-owned health facilities include 18 PHC centres and the Lagos State University Teaching Hospital (LASUTH). There is no secondary

health care facility in Ikeja LGA as the existing one was upgraded to become the teaching hospital (LASUTH). Alimosho LGA is the largest in the state and has an estimated population of 1,469,134 residents. Health services are also delivered through a mix of government-owned and private health facilities. The LGA hosts a government-owned secondary health facility, Alimosho General Hospital, Igando, and 33 PHCs.

## Trial design

A before and after quasi-experimental study approach was used to conduct the study amongst HCWs and their clients (mothers of under-five children). The study had three phases: baseline, intervention, and post-intervention. The intervention phase was designed to meet identified gaps from the baseline phase.

## Sample size

The sample size was calculated using the appropriate formula for sample size calculation for intervention studies [17]. A minimum of 100 health care workers (50 in each LGA) and 200 community members (100 in each LGA) were required to achieve the study objectives.

## Participants eligibility criteria

All health care workers above 18 years involved in patient care who were willing to participate in all phases of the study but PHC workers not willing to give consent were excluded. The clients were mothers of under-five children utilizing the PHCs who were willing to participate in all phases of the study.

## Sampling techniques

A multi-stage sampling method was used to recruit participants for this study. A simple random sampling method was used to select two out of 20 LGAs in the state. The selected LGAs were Ikeja and Alimosho. PHCs were selected based on having a high patient load (four facilities in Ikeja LGA and five facilities in Alimosho LGA) and all eligible health workers in the selected facilities were recruited into the study. These PHCs were selected by simple random sampling. Eligible clients were recruited consecutively.

## Survey instruments

Two almost identical study instruments were used to obtain data. The first was a PHC workers questionnaire and the other was a client questionnaire. Both were designed to elicit the knowledge, attitudes, and preventive practices of COVID-19 among the respondents. The questionnaires were developed based on current literature on the subject [18, 19].

The questionnaires consisted of three parts. The first part collected information on the biodata and work experience of the health workers including whether they had attended courses on COVID-19 and infection prevention and control and their sources of information on COVID-19. The second part which consisted of 31 questions was further sub-divided into 3 parts consisting of "yes, No and don't know options on their knowledge of COVID-19. The knowledge questions (n = 11) were on the causative agent, incubation period, symptoms, mode of spread, and severity of COVID-19. Attitude questions (n = 9) were responses to the likelihood if the participants had COID-19 and to others suspected of having the infection. The attitudes were assessed using a 5-point Likert Scale: "strongly disagree, disagree, neutral, agree, and strongly agree". Preventive practice questions (n = 9) elicited information on what respondents did to prevent the disease such as hand washing, use of face masks, and physical

distancing. The preventive practices domain was assessed using a 4-point Likert Scale: "always, often, sometimes and never". The third part consisting of 6 questions was on vaccine acceptability.

Face and content validity was done by the research team. The reliability coefficients of the HCWs tool sub-scales were: knowledge, $\alpha = 0.63$; attitude, $\alpha = 0.60$; and practice, $\alpha = 0.86$. The reliability coefficients for the clients' tool were: knowledge, $\alpha = 0.82$; attitude, $\alpha = 0.67$; and practice, $\alpha = 0.80$. Pre-testing of the study instruments was done amongst 20 HCWs in an LGA not involved in the study and who met the eligibility requirements of the study. This was done to test the flow of questions, and clarity of the questions to the interviewer.

## Conduct of the study and intervention

The project was carried out in three phases: baseline (pre-intervention), intervention, and post-intervention phases. The baseline survey was to determine the level of COVID-19 knowledge, attitude, and practices among health workers using the pre-tested self-administered questionnaire. Questionnaires were administered to clients on their first visit to the PHC for childcare services to assess their knowledge, attitudes, and preventive practices against COVID-19. The tool for the client survey was interviewer-administered and collected on the kobo collect form electronically. The baseline study lasted for four weeks.

The intervention was developed based on the theory of change model a concept that is used to explain how and why change occurs [20]. There are several approaches to the use of the theory. It is a process that focuses on the need, the assumptions, the desired interventions, and the changes desired. For the study, the problem was inadequate knowledge of COVID-19 amongst the HCWs and clients. The assumptions were willingness to learn and previous knowledge of the HCWs. The key constructs were root cause analysis to identify the gaps; input which is the web-based training programmes; the impact (study outcome) was the increased knowledge amongst HCWs and clients and some measure of behaviour change. The long-term outcome constructs were sustained behaviour change and reduced transmission of COVID-19 in the community would. occur beyond the duration of the study. Development of the application and the modules was done over three months.

The intervention was a virtual training on COVID-19 for PHC workers. A website was developed for the training. The website was registered, and the name was approved by the regulatory authorities. The website address was www.healthworkerstraining.com.ng. The training was conducted to improve the knowledge of the PHC workers on COVID-19 and meet the gaps identified from the baseline survey. The website contained videos, presentation slides, quizzes, case scenarios, and problem-solving sessions with provisions for discussion and comments.

There were five main modules in addition to an introductory session. Each module lasted about 25 minutes. The modules were: epidemiology, risk communication, community engagement, prevention of COVID-19, and vaccination against COVID-19. The delivery methods consisted of a mixture of lectures, practical demonstrations, assignments, and question, and answer sessions. Each module had pre-test and post-test multiple-choice questions (one in four best answer options) to monitor participants' progress. The training was self-paced and delivered entirely online. Four weeks were provided to enable health workers complete the training and the post-intervention data collection was done over one week. After the training, the same set of clients were exposed to the trained PHC workers for one-on one counselling on invitation to the PHC. Each session lasted for about ten minutes. The clients were invited within one month of completion of the training by the health workers.

Six research assistants with a minimum of Ordinary National Diploma (OND) qualification, fluent in both Yoruba and English languages were trained for two days on the aim and

objectives of the study, sampling procedures, administration of the questionnaire, with practical sessions.

## Study outcomes

The study outcomes were changes in the knowledge, attitudes, and preventive practices which were scored. For the knowledge domain, each correct answer was assigned one mark and a wrong answer zero. The maximum knowledge score of HCWs was 33. The attitude of the HCWs was scored, the range was from one to five, and the maximum score was 45. The preventive practices were scored, the range from one to four, and the maximum score for the preventive practice domain was 36. Categorization of knowledge, attitudes and preventive practices was done into good if the scores were at least 66% [21]. Scoring for the clients was like that of the HCWs except that the maximum knowledge score was 27 and the attitude score was 40.

## Data management

Data were entered and cleaned on Microsoft Excel and imported to STATA where it was analyzed using STATA SE Version 12. Changes in knowledge, attitudes and preventive practices of the HCWs and the clients were compared pre-intervention and post-intervention using paired t-test (when the data were normally distributed) and Wilcoxon signed-rank test (when the data were not normally distributed). Associations between the socio-demographic characteristics of respondents and their post-intervention COVID-19 knowledge, attitude, and preventive practices were assessed using an independent sample t-test and one-way analysis of variance (ANOVA) when data were normally distributed or Wilcoxon signed-rank and Kruskal-Wallis equality-of-populations rank test as appropriate when the data were not normally distributed. The level of significance was set at $P < 0.05$.

## Ethical considerations

The study was conducted in compliance with Good Clinical Practice standards. Ethical approval was obtained from the Health Research Ethics Committee of the Lagos State University Teaching Hospital (REFERENCE NO: LREC/06/10/1501). Written informed consent was obtained from all the participants after explaining the study's aim and objectives.

## Results

### PHC workers survey

Sixty-three primary healthcare workers who completed the virtual training and post-training assessment were interviewed after and had post-training interaction with clients enrolled for this study out of the 100 that were enrolled at baseline giving a 63% completion rate (Fig 1).

The socio-demographic characteristics of the HCWs are shown in Table 1. The mean age at the end of the study was 39.2±9.9 years respectively. Females constituted 85% of the group. Half of the HCWS were nurses. The mean years of work experience at the end of the study was 12.5±10.0 years. Table 2 displays information about awareness and knowledge of the HCWs on the disease pre- and post-training. About 95.2% had heard of the Covid-19 virus during the baseline study and all post-training. About 87% at baseline and 98.4% at post-training knew that the incubation period was 1–14 days. The level of knowledge was high on symptoms (> 90%), modes of transmission (>80%), and preventive measures (>93%). Sixty -two (98.4%) of the HCWs were willing to take the COVID-19 vaccine at baseline and all were willing at the end of the study.

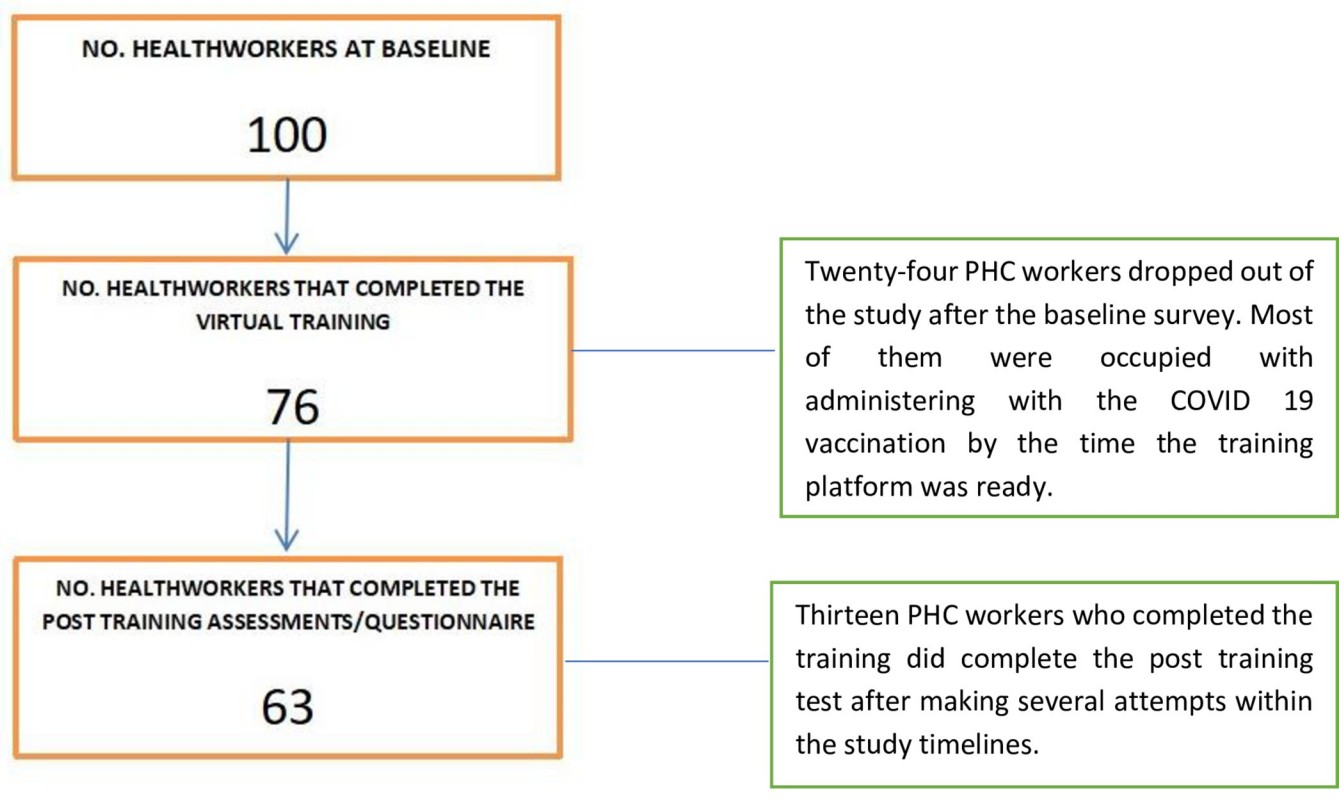

**Fig 1. Study flow chart for PHC workers.**

There were no statistically significant differences in the mean COVID 19 knowledge, attitude, and preventive practices scores of PHC workers before and after training, even though they had higher scores (see Table 3). Table 4 shows that health workers more than 30 years had a significantly higher mean knowledge score (28.80± 1.71) than those who were 30 years or less (27.81±2.64). Married PHC workers had higher mean knowledge scores compared with the single. PHC workers in Alimosho LGA had higher mean knowledge scores than those in Ikeja LGA. The preventive practice scores of the HCWs were not significantly associated with any attributes of the participants as shown in Table 5.

## Client's survey

One hundred and thirty-three clients completed the study and were interviewed after their interactions with trained PHC workers out of the 226 that were surveyed at baseline, giving a completion rate of 58.9% (Fig 2).

The mean age of clients at the end of the study was 30.9 ± 5.0 years. The majority (78%) earned ₦50,000 or less and 50% had a secondary level of education (Table 6). The awareness and knowledge of the clients are shown in Table 7. At the baseline study, 98.5% of clients had heard of the Covid-19 virus and among those who have heard, 80.2% stated that the Covid-19 virus was highly infectious, and 73% knew that the virus can be transmitted by asymptomatic patients. The interaction with trained PHC workers led to an increase in clients' knowledge of symptoms (> 80%), modes of transmission (> 80%) and preventive measures (> 90%). The willingness to accept COVID-19 vaccines amongst the clients increased from 60 (45.1%) to 92 (69.2%) after meeting with trained HCWs.

**Table 1. Socio-demographic characteristics of PHC workers.**

| Socio-Demographic Characteristics | Post-training |
|---|---|
| | **(n = 63)** |
| | **N (%)** |
| **LGA** | |
| Ikeja | 39(61.9) |
| Alimosho | 24(38.1) |
| **Age-group in years** | |
| ≤30 | 14 (22.2) |
| >30 | 49 (77.8) |
| **Mean (SD)** | 39.2±9.9 |
| **Gender** | |
| Female | 54(85.7) |
| Male | 9(14.3) |
| **Years of working experience** | |
| **Mean (SD)** | 12.5±10.0 |
| ≤10 | 34(54.0) |
| >10 | 29(46.0) |
| **Marital Status** | |
| Married | 47(74.6) |
| Single | 16(25.4) |
| **Participated in the IPC training** | 57(90.5) |
| **Profession** | |
| Doctor | 8(12.7) |
| Nurse | 33(52.4) |
| CHEW | 11(17.4) |
| Community Health officer | 9(14.3) |
| Laboratory scientist | 1(1.6) |
| Volunteer health worker | 1(1.6) |

Table 8 shows that there were significantly higher scores in the knowledge, attitude, and preventive practices scores of clients after interaction with trained PHC workers (p<0.001). Single clients had significantly higher mean knowledge scores than the married but clients in Ikeja LGA had significantly higher knowledge scores than those in Alimosho LGA after meeting with trained PHC workers. (p<0.05), Table 9. Table 10 displays information about the association between preventive practice scores and clients' sociodemographic characteristics. Clients attending PHCs in Alimosho LGA had statistically significant better practices compared to those attending PHCs in Ikeja LGA.

## Discussion

This intervention study set out to determine the knowledge of, attitudes to, and preventive practices of HCWs and their clients at baseline and to evaluate the effect of the training in both groups.

At baseline, HCWs' knowledge was high, attitudes were positive and preventive practices were good. This high level of knowledge agrees with high levels reported from the UAE [3], Uganda [6], Ethiopia [7], and Nigeria [8, 9]. It is higher than the levels reported from Yemen [4], and Mozambique [5]. Nine out of ten HCWs correctly identified the common symptoms such as shortness of breath and cough. The high level of knowledge is probably a reflection of

**Table 2. Awareness and knowledge of COVID-19 among PHC workers pre-and post-training.**

| Awareness and knowledge of COVID-19 | Pre-test (n = 63) N (%) | Post-test (n = 63) N (%) |
|---|---|---|
| Ever heard of the COVID-19 virus | 60(95.2) | 63(100) |
| **Awareness that COVID-19 is real** | 60(95.2) | 62(98.4) |
| SARS-COV-2 is the virus that causes Covid-19 | 52(82.5) | 51(81.0) |
| It's highly infectious | 59(93.6) | 62(98.4) |
| The incubation time is 1–14 days | 52(82.5) | 62(98.4) |
| The asymptomatic patient can transmit the virus | 59(93.6) | 62(98.4) |
| **Knowledge of Symptoms** | | |
| Sore throat | 58(92.1) | 62(98.4) |
| Cough | 58(92.1) | 62(98.4) |
| Bleeding nose | 18(28.5) | 14(22.2) |
| Loss of taste and smell | 58(92.1) | 61(96.8) |
| Shortness of breath | 59(93.6) | 63(100.0) |
| Fever | 58(92.1) | 62(98.4) |
| Joint pain | 42(66.7) | 48(76.2) |
| Diarrhoea | 36(57.1) | 34(54.0) |
| Rash | 17 (27.0) | 4(6.3) |
| Abdominal pain | 31(49.2) | 27(42.9) |
| **Knowledge of Mode of Transmission** | | |
| Spread through contact with an infected person | 49(77.7) | 51(81.0) |
| Spread through contact with an infected surface | 56(88.8) | 60(95.2) |
| Sneezing | 59(93.6) | 62(98.4) |
| Talking | 49(77.7) | 51(81.0) |
| Spread through coughing | 59(93.6) | 63(100.0) |
| **Knowledge of Means of Prevention** | | |
| Hand washing | 60(95.2) | 63(100.0) |
| Alcohol-based hand sanitizer | 60(95.2) | 63(100.0) |
| Cannot be prevented by alcohol intake | 52(82.5) | 61(96.8) |
| Covid-19 vaccine | 59(93.6) | 61(96.8) |
| Cough hygiene | 58(92.1) | 59(93.7) |
| Social distancing | 60(95.2) | 61(97.0) |
| Face mask | 60(95.2)) | 63(100.0) |
| Covid-19 may lead to serious complication | 57 (90.1) | 59(93.7) |
| **Knowledge of Population at Risk of Poor Outcomes from Covid-19** | | |
| Children | 35(55.6) | 28(44.4) |
| Pregnant women | 49(77.7) | 46(73.0) |
| Elderly | 59(93.6) | 62(98.4) |
| People with underlying medical conditions | 60(95.2) | 63(100.0) |
| A suspected case can only be ruled after 2 consecutive negative tests | 49(77.7) | 53(84.1) |

the multiple sources of information and training received by the HCWs since the pandemic began. At the time of the study, the pandemic had lasted for over a year in Nigeria, thus information and awareness on COVID-19 were abundant. Therefore, it was not surprising that HCWs had a high level of knowledge. Had the study been conducted very early during the pandemic, perhaps the level of knowledge may not have been as high. This high level of knowledge gives assurance that the health workers were more likely to give correct information to their clients and help further in changing their behaviour. Nevertheless, there were some gaps

**Table 3. Pre-and post-training knowledge attitude and preventive practices scores of PHC workers.**

| Variable domain | Baseline | Post-training | Statistics | P-value |
|---|---|---|---|---|
| **Knowledge Score** Mean ± SD | 28.48 ± 2.24 | 28.52 ± 1.92 | 0.14** | 0.446 |
| **Attitude Score** Mean ± SD | 34.44 ± 3.72 | 34.83 ± 3.84 | 0.63** | 0.267 |
| **Practice Score** Mean ± SD | 33.95 ± 3.13 | 34.60 ± 2.73 | 1.81* | 0.070 |

*Wilcoxon signed-rank test

**paired T-test.

**Table 4. Association between PHC workers' socio-demographic characteristics and COVID-19 knowledge scores post-training.**

| Variables | Post Training Knowledge Score Mean ± SD | Test statistics | *P*-value |
|---|---|---|---|
| **Age group (years)** | | | |
| ≤30 | 27.57 ± 2.34 | -2.17^ | 0.034* |
| >30 | 28.80 ± 1.71 | | |
| **Gender** | | | |
| Female | 28.52 ± 1.98 | -0.05^ | 0.479 |
| Male | 28.56 ± 1.59 | | |
| **Years of Experience** | | | |
| ≤10 | 28.44 ± 1.86 | -0.37^ | 0.357 |
| >10 | 28.62 ± 2.01 | | |
| **Marital status** | | | |
| Married | 28.77 ± 1.56 | 1.75^ | **0.043** |
| Single | 27.81 ± 2.64 | | |
| **Participated in any IPC training** | | | |
| Yes | 28.52 ± 1.96 | 1.19^ | 0.425 |
| No | 28.67 ± 1.63 | | |
| **Received online training on Covid-19** | | | |
| Yes | 28.47 ± 2.05 | 0.39^ | 0.348 |
| No | 28.69 ± 1.49 | | |
| **LGA** | | | |
| Alimosho | 29.13 ± 1.70 | 2.00^ | **0.025** |
| Ikeja | 28.15 ± 1.97 | | |
| **Level of Education** | | | |
| Diploma | 28.45 ± 2.42 | 0.56^^ | 0.574 |
| Bachelor's degree | 28.44 ± 1.54 | | |
| Master's degree | 29.40 ± 2.07 | | |
| **Profession** | | | |
| Doctor | 28.63 ± 2.50 | 0.27^^ | 0.847 |
| Nurse | 28.55 ± 1.44 | | |
| CHEW, Lab scientist & volunteers | 28.15 ± 2.51 | | |
| Community health officer | 28.89 ± 2.20 | | |

^ Independent T-test

^^ One-way ANOVA.

**Table 5. Association between the socio-demographic characteristics of the health workers and COVID-19 prevention practice scores.**

| Variables | Post Training Practices Score Mean ± SD | Test statistics | *P*-value |
|---|---|---|---|
| **Age group (years)** | | | |
| ≤30 | 34.71 ± 3.15 | 0.00! | 1.000 |
| >30 | 34.57 ± 2.64 | | |
| **Gender** | | | |
| Female | 34.78 ± 2.41 | 0.62! | 0.535 |
| Male | 33.56 ± 4.25 | | |
| **Years of Experience** | | | |
| ≤10 | 34.71 ± 2.86 | 1.27! | 0.203 |
| >10 | 34.48 ± 2.63 | | |
| **Marital status** | | | |
| Married | 34.55 ± 2.68 | 0.13! | 0.899 |
| Single | 34.75 ± 3.00 | | |
| **Participated in any IPC training** | | | |
| Yes | 34.67 ± 2.71 | -0.78! | 0.436 |
| No | 34.00 ± 3.10 | | |
| **Received online training on Covid-19** | | | |
| Yes | 34.57 ± 2.91 | -0.73! | 0.469 |
| No | 34.69 ± 2.21 | | |
| **LGA** | | | |
| Alimosho | 35.00 ± 1.74 | -0.08! | 0.935 |
| Ikeja | 34.36 ± 3.19 | | |
| **Level of Education** | | | |
| Diploma | 34.77 ± 2.52 | 1.33!! | 0.514 |
| Bachelor's degree | 34.33 ± 3.01 | | |
| Master's degree | 35.80 ± 0.45 | | |
| **Profession** | | | |
| Doctor | 34.38 ± 4.21 | 2.17!! | 0.539 |
| Nurse | 34.39 ± 2.68 | | |
| CHEW, Lab scientists and volunteers | 35.23 ± 1.96 | | |
| Community health officer | 34.67 ± 2.65 | | |

! Two-sample Wilcoxon rank-sum (Mann-Whitney) test!! Kruskal-Walli's equality-of-populations rank test.

in the knowledge of the epidemiology of the disease amongst the HCWs which formed the basis for the intervention

The clients' levels of knowledge at baseline were fair to moderate. Common symptoms of the disease such as fever, cough and shortness of breath were known although knowledge of more specific ones like anosmia and ageusia was lower. The level of knowledge of the clients is comparable with studies in Pakistan [22], but much higher than was reported from Kano, northern Nigeria [12], which may be due to lower access to information and lower levels of literacy in Kano. The proportion of clients in this study who were aware that an asymptomatic person could transmit COVID-19 was similar to a study amongst mothers of under-five children in Enugu [13]. In addition, the 70% rate of good preventive practices found at baseline was much higher than the 49% reported amongst pregnant women in Northern Ghana [14]. The vaccine uptake was low probably because of the non-availability of vaccines in the state at the time of the baseline study.

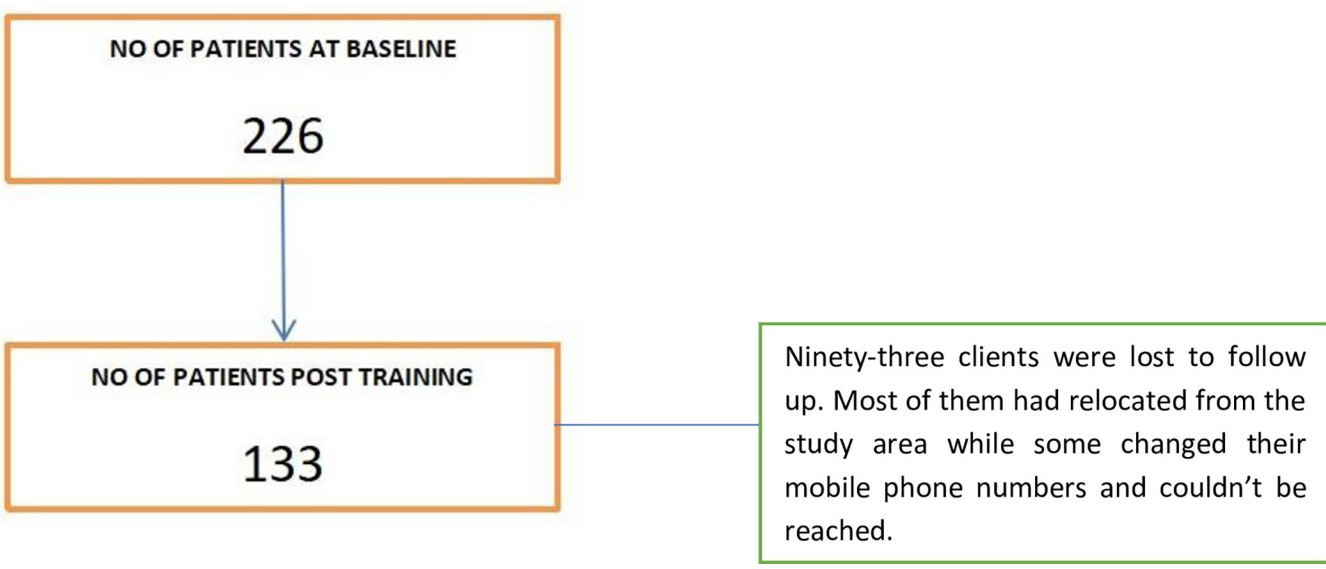

**Fig 2. Study flow chart for clients.**

The training produced changes in all the domains of knowledge, attitudes, and preventive practices of the HCWs. There was a higher level of knowledge among the health workers in all areas; be it in the knowledge of symptoms (above 90%), mode of transmission and preventive measures. Vaccine uptake was observed to have increased within the period although it cannot be said to be due directly to the intervention. The training was effective in producing the desired change and justifies the use of the theory of change model as a conceptual framework for the intervention. The improvement in the HCWs is in conformity with an intervention reported in Pakistan, where a randomized trial amongst nurses in Pakistan utilized WhatsApp to deliver an educational intervention for 12 weeks reported that the intervention group showed significantly better results in infection prevention control and leadership and communication [23]. Further-more, the deployment of a COVID-19 training app amongst a group of HCWs in Nigeria led to a significant increase in knowledge from pre-test scores of 54% to 74% at the post-test in confor-mity with the findings of the present study [15]. These studies [15, 23], and ours show the value of the effectiveness of online training in building the capacity of HCWs.

Older HCWs had higher knowledge scores than younger ones in contrast to findings from Ethiopia [7], and Nigeria [15]. We posit that the older HCWs were more experienced than their younger colleagues and not just due to cognitive abilities alone. In addition, it may be that the age group category used in our study (30 years) included many more brilliant but young health care workers. The study did not find any association between knowledge and professional cadre of the HCWs unlike what was found in the UAE [3], and Nigeria [9], and this is probably because of the small sample size in our study. The higher levels of knowledge in Alimosho LGAs and the influence of marital status cannot be easily accounted for. We also did not find any significant association between HCWs' socio-demographic characteristics and preventive practices unlike a study from Ethiopia [7], perhaps on account of the small sample size.

Far more important than the changes observed in the HCWs is the effect of the training on the clients. Contact of the clients with HCWs who had received the intervention led to higher levels of knowledge and better recognition in all domains of knowledge, attitudes and preven-tive practices examined. There was better recognition of symptoms and of correct preventive practices. Besides, the proportion of those who had received the COVID-19 vaccine rose from

**Table 6. Socio-demographic characteristics of clients.**

| Socio-Demographic Characteristics | Post-training |
|---|---|
| | (n = 133) |
| | N (%) |
| **LGA** | |
| Ikeja | 75(56.4) |
| Alimosho | 58(41.4) |
| **Age-group in years** | |
| **Mean (SD)** | 30.9±5.0 |
| ≤30 | 66(49.6) |
| >30 | 67(50.4) |
| **Religion** | |
| Christian | 92(69.2) |
| Muslim | 41(30.8) |
| **Level of Education** | |
| None | 0(0.0) |
| Primary | 4(3.0) |
| Secondary | 66(49.6) |
| Tertiary | 63(47.4) |
| **Marital Status** | |
| Married | 130(97.7) |
| Single | 3(2.3) |
| **Ethnicity** | |
| Yoruba | 76(57.1) |
| Igbo | 33(24.8) |
| Hausa | 2(1.5) |
| Others | 22(16.5) |
| **Average Monthly Income** | |
| **Mean (SD)** | |
| ≤ ₦50,000 | 103(77.4) |
| >₦50,000 | 28(21.1) |
| Non -Response | 2(1.5) |
| **Employment Status** | |
| Employed | 117(87.9) |
| Unemployed | 16 (12.0) |

less than half to about seven out of ten and were willing to recommend the vaccine to others, although not entirely due to the intervention. The changes confirm the efficacy of the intervention and its benefit not only to the health worker but to their clients. This efficacy of the intervention is like what was seen amongst community members in Iran [18], and Thailand [24]. The finding that single clients had a higher level of knowledge cannot be accounted for. The benefit of clients receiving correct information cannot be overemphasised in the light of conspiracy theories. Contact with health workers who are knowledgeable builds confidence and may lead to sustained utilization of PHCs.

The efficacy of the intervention is attributable in part to the expertise of the research team, the authentic information in the training materials, the method of delivery, and the willingness of the HCWs to participate on account of their perceived benefits. Virtual training such as this we undertook has numerous advantages. They are low-cost, non-personal, allow for learning at the individual level and pace, are available for continuous learning. They are invaluable for

**Table 7. Client's awareness and knowledge of Covid-19 before and after interaction with trained PHC workers.**

| Awareness and knowledge of covid-19 | Before | After |
|---|---|---|
| | (n = 133) | (n = 133) |
| | N (%) | N (%) |
| Ever heard of the Covid-19 virus | 131 (98.5) | 132(99.2) |
| **Source of Information** | **(n = 131)** | **(n = 132)** |
| Media | 125(95.4) | 130(98.5) |
| Religious centres | 58(44.3) | 128(97.0) |
| Official website | 31(23.7) | 75(56.8) |
| **Awareness that Covid-19 is real** | 106(80.9) | 131(99.2) |
| It's highly infectious | 105(80.2) | 127(96.2) |
| An asymptomatic patient can transmit the virus | 95(72.5) | 120(90.9) |
| **Knowledge of Symptoms** | | |
| Sore throat | 108(82.4) | 124(93.9) |
| Cough | 122(93.1) | 130(98.5) |
| Bleeding nose | 73(55.7) | 73(55.3) |
| Loss of taste | 90(68.7) | 114(86.4) |
| Shortness of breath | 112 (85.5) | 123(93.2) |
| Loss of smell | 83(63.4) | 112(84.8) |
| Fever | 114(87.0) | 129(97.7) |
| Joint pain | 46 (35.1) | 74(56.1) |
| Diarrhoea | 54(41.2) | 96(72.7) |
| Rash | 39(29.8) | 26(19.7) |
| Abdominal pain | 48(36.6) | 52 (39.4) |
| **Knowledge of Mode of Transmission** | | |
| Spread through contact with an infected person | 117(89.3) | 128(97.0) |
| Spread through contact with an infected surface | 122(93.1) | 126(95.5) |
| Sneezing | 126(96.2) | 131(99.2) |
| Talking | 110(84.0) | 110(83.3) |
| Spread through coughing | 125(95.4) | 132(100.0) |
| **Knowledge of Means of Prevention** | | |
| Staying at home | 124 (94.7) | 122(92.4) |
| Hand washing | 129(98.5) | 131(99.2) |
| Alcohol-based hand sanitizer | 129(98.5) | 130(98.5) |
| Cannot be prevented by alcohol intake | 48(36.6) | 98(74.2) |
| Covid-19 vaccine | 85(64.9) | 123(93.2) |
| Cough hygiene | 129(98.5) | 130(98.5) |
| Social distancing | 129(98.5) | 132(100.0) |
| Face mask | 130(99.2) | 132(100.0) |
| Covid-19 may lead to serious complication | 116 (88.5) | 126(95.5) |
| Doctor/nurse talked about Covid-19 outbreak | 95(72.5) | 130(98.5) |

social distancing and for dealing with large groups. This is probably the way to adopt in this era of the new normal.

## Conclusion

This study has shown that virtual training is an effective way to improve the knowledge and skills of health workers who can then impact their clients and bring immense benefit to them, especially in the control of pandemic prone diseases.

**Table 8. Knowledge, attitude, and preventive practices of clients before and after interaction with trained PHC workers.**

| Mean ± SD | Baseline | Post-training | Test statistics | *P*-value |
|---|---|---|---|---|
| **Knowledge Score** | 20.00± 4.25 | 22.96 ± 2.32 | -7.03* | <0.001 |
| **Attitude Score** | 26.50 ± 4.01 | 28.74 ± 5.20 | 4.63** | <0.001 |
| **Practice Score** | 26.39 ± 5.28 | 28.47± 4.61 | 3.85** | <0.001 |

*Wilcoxon signed-rank test.

** Paired T test.

# Limitations of the study

The positive effect of the intervention amongst HCWs and clients cannot be attributable solely to the intervention as both groups were exposed to other sources of information although not measured which could have contributed to the changes. Mothers were the clients in the study as they were more readily available at PHCs more than men and this limits the generalization of the changes mainly to mothers of children under-five years. The sample size for the study

**Table 9. Association between the socio-demographic characteristics of clients and knowledge scores after interaction with trained PHC workers.**

| Sociodemographic characteristics | After intervention (Mean ± SD) | Test Statistics | *P*-value |
|---|---|---|---|
| **Age group (years)** | | | |
| ≤30 | 23.08± 2.52 | -0.92! | 0.359 |
| >30 | 22.85± 2.11 | | |
| **Religion** | | | |
| Christian | 23.21 ± 2.19 | 1.70! | 0.090 |
| Muslim | 22.41 ± 2.54 | | |
| **Marital status** | | | |
| Married | 22.91 ± 2.31 | -2.07! | **0.038** |
| Single | 25.33 ± 1.15 | | |
| **Ethnicity** | | | |
| Yoruba | 22.62 ± 2.48 | 3.73!! | 0.292 |
| Igbo | 23.39± 2.24 | | |
| Hausa | 22.50 ± 0.71 | | |
| Others | 23.55 ± 1.77 | | |
| **Monthly Income** | | | |
| ≤₦50,000 | 22.89 ± 2.27 | 1.23! | 0.218 |
| >₦50,000 | 23.32 ± 2.54 | | |
| **Employment Status** | | | |
| Employed | 22.97 ± 2.35 | 0.28! | 0.779 |
| Unemployed | 22.88 ± 2.16 | | |
| **LGA** | | | |
| Alimosho | 22.31± 2.71 | -2.45! | **0.014** |
| Ikeja | 23.47 ± 1.83 | | |
| **Level of education** | | | |
| Primary | 21.75 ± 2.06 | 2.05! | 0.360 |
| Secondary | 23.09 ± 2.38 | | |
| Tertiary | 22.90 ± 2.28 | | |

**!** Two-sample Wilcoxon rank-sum (Mann-Whitney) test.

**!!** Kruskal-Walli's equality-of-populations rank test.

**Table 10. Association between the socio-demographic characteristics of clients and preventive practice scores after interaction with trained PHC workers.**

| Variables | Post-test score (Mean ± SD) | Test statistics | *P*-value |
|---|---|---|---|
| **Age group (years)** | | | |
| ≤30 | 27.92 ± 4.06 | 1.37^ | 0.087 |
| >30 | 29.01 ± 5.07 | | |
| **Religion** | | | |
| Christian | 28.09 ± 4.61 | 1.45^ | 0.074 |
| Muslim | 29.34 ± 4.54 | | |
| **Marital status** | | | |
| Married | 28.38 ± 4.56 | 1.60^ | 0.056 |
| Single | 32.67 ± 5.77 | | |
| **Ethnicity** | | | |
| Yoruba | 28.68 ± 4.61 | 1.23^^ | 0.302 |
| Igbo | 28.91 ± 4.88 | | |
| Hausa | 23.50 ± 3.54 | | |
| Others | 27.54 ± 4.15 | | |
| **Monthly Income** | | | |
| ≤₦50,000 | 28.33 ± 4.48 | 0.67^ | 0.507 |
| >₦50,000 | 29.00 ± 5.28 | | |
| **Employment Status** | | | |
| Employed | 28.43 ± 4.59 | 0.31^ | 0.755 |
| Unemployed | 28.81 ± 4.92 | | |
| **LGA** | | | |
| Alimosho | 29.59 ± 4.57 | 2.49^ | **0.014** |
| Ikeja | 27.61 ± 4.48 | | |
| **Level of education** | | | |
| Primary | 27.00 ± 4.08 | 1.79^^ | 0.170 |
| Secondary | 27.82 ± 4.75 | | |
| Tertiary | 29.25 ± 4.42 | | |

^ Independent t-test.

^^ One-way ANOVA.

was small. This was in part because the study was meant to demonstrate proof of concept. In addition, the response rate was not as high as expected. However, this is not unusual in studies amongst health workers [5, 15], like the participants in this study who for various reasons could not complete the post-intervention assessment. The lack of a control group is another limitation of the study.

## Recommendations

We recommend the adoption of virtual training to improve the knowledge and skills of HCWs at the first level of care. Such online training should be the new norm and method of continuing professional development. Web-based apps are invaluable, and priority should be given to apps that do not require a lot of data. They should be usable on android phones which are used by more people. Social media for example WhatsApp can be used for educational interventions. Collaboration with the information and technology sector along with telecommunication companies is essential and should be sought.

## Supporting information

**S1 File. LASRIC data file.**
(XLSX)

## Acknowledgments

We are grateful for the support provided by the Government of Lagos State, the leadership of the Lagos State Research and Innovation Council (LASRIC), the leadership of Lagos State University, the Medical Officers of Health of Ikeja and Alimosho LGA, the health care workers and the mothers of under-five children who participated in the study.

## Author Contributions

**Conceptualization:** Olumuyiwa O. Odusanya.

**Data curation:** Olumuyiwa O. Odusanya, Adeyinka Adeniran, Omowunmi Q. Bakare, Babatunde A. Odugbemi.

**Formal analysis:** Olumuyiwa O. Odusanya, Adeyinka Adeniran, Omowunmi Q. Bakare, Babatunde A. Odugbemi.

**Funding acquisition:** Olumuyiwa O. Odusanya.

**Investigation:** Olumuyiwa O. Odusanya, Adeyinka Adeniran, Omowunmi Q. Bakare, Babatunde A. Odugbemi, Oluwatoyin A. Enikuomehin, Olugbenja O. Jeje, Angela C. Emechebe.

**Methodology:** Olumuyiwa O. Odusanya, Adeyinka Adeniran, Omowunmi Q. Bakare, Babatunde A. Odugbemi.

**Project administration:** Olumuyiwa O. Odusanya.

**Software:** Oluwatoyin A. Enikuomehin.

**Supervision:** Olumuyiwa O. Odusanya, Adeyinka Adeniran, Omowunmi Q. Bakare, Oluwatoyin A. Enikuomehin, Olugbenja O. Jeje, Angela C. Emechebe.

**Validation:** Adeyinka Adeniran, Omowunmi Q. Bakare, Babatunde A. Odugbemi.

**Writing – original draft:** Olumuyiwa O. Odusanya, Adeyinka Adeniran, Omowunmi Q. Bakare, Babatunde A. Odugbemi.

**Writing – review & editing:** Olumuyiwa O. Odusanya, Adeyinka Adeniran, Babatunde A. Odugbemi, Oluwatoyin A. Enikuomehin, Olugbenja O. Jeje, Angela C. Emechebe.

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
