## [Decision Letter · Decision Letter 0]

22 Aug 2022

PONE-D-22-16429The Editor-in-Chief PLoS One

FULL TITLE

Building capacity of Primary Health Care workers and Clients on COVID-19: Results from a web-based trainingPLOS ONE

Dear Dr. Olumuyiwa Odusanya,

Thank you for submitting your manuscript to PLOS ONE. After careful consideration, we feel that it has merit but does not fully meet PLOS ONE’s publication criteria as it currently stands. Therefore, we invite you to submit a revised version of the manuscript that addresses the points raised during the review process. The topic is apt and the research question, clear. The manuscript is fairly well written but requires specific technical details in the methods and results to improve the quality of the paper. Kindly respond to all the issues raised by both reviewers.

Please submit your revised manuscript within Oct 06 2022 11:59PM. If you will need more time than this to complete your revisions, please reply to this message or contact the journal office at plosone@plos.org. Please include the following items when submitting your revised manuscript:A rebuttal letter that responds to each point raised by the academic editor and reviewer(s). You should upload this letter as a separate file labeled 'Response to Reviewers'.A marked-up copy of your manuscript that highlights changes made to the original version. You should upload this as a separate file labeled 'Revised Manuscript with Track Changes'.An unmarked version of your revised paper without tracked changes. You should upload this as a separate file labeled 'Manuscript'.If applicable, we recommend that you deposit your laboratory protocols in protocols.io to enhance the reproducibility of your results. Protocols.io assigns your protocol its own identifier (DOI) so that it can be cited independently in the future. For instructions see: https://journals.plos.org/plosone/s/submission-guidelines#loc-laboratory-protocols. Additionally, PLOS ONE offers an option for publishing peer-reviewed Lab Protocol articles, which describe protocols hosted on protocols.io. Read more information on sharing protocols at https://plos.org/protocols?utm_medium=editorial-email&utm_source=authorletters&utm_campaign=protocols.

We look forward to receiving your revised manuscript.

Kind regards,

Ogochukwu Chinedum Okoye

Academic Editor

PLOS ONE

Journal Requirements:

2. PLOS ONE does not copy edit accepted manuscripts (https://journals.plos.org/plosone/s/criteria-for-publication#loc-5). To that effect, please ensure that your submission is free of typos and grammatical errors.  

Reviewers' comments:

Reviewer's Responses to Questions

**Comments to the Author**

1. Is the manuscript technically sound, and do the data support the conclusions?

Reviewer #1: Yes

Reviewer #2: Yes

2. Has the statistical analysis been performed appropriately and rigorously? 

Reviewer #1: Yes

Reviewer #2: Yes

3. Have the authors made all data underlying the findings in their manuscript fully available?

Reviewer #1: Yes

Reviewer #2: Yes

4. Is the manuscript presented in an intelligible fashion and written in standard English?

Reviewer #1: Yes

Reviewer #2: Yes

5. Review Comments to the Author

Reviewer #1: 1. The manuscript is technically sound with regards to its research question of assessing the effect of web based training on health care workers and their clients on COVID 19.

2. The statistical analysis in general is appropriate and rigorous. However one is curious to know why an "independent t-test" was used to measure differences in knowledge, attitude and practice (KAP) pre- and post- intervention instead of a "paired t-test"?

3. The Journal editor will be in the best position to answer that, as this reviewer is not privy to the data. Based on PLOS ONE requirements i however assume a YES.

4.The manuscript is presented in an intelligible fashion and written in standard English.

Reviewer #2: Title – Building capacity of Primary Health Care workers and Clients on COVID-19: Results

3 from a web-based training

The topic appropriate for the study

Abstract – Structured and well written

Background –

Detailed information on statement of problem as well as rational for the study clearly presented.

Line 89 – 90: Almost 80% had positive - had positive what?

82% used face masks - desirable to know what use of face mask means in the study. Is it consistent use of face mask or ever use of face mask?

Methods – Well described in details

Line 127 – 128: The population is estimated to be over 20 million.- Provide reference

The health care of its residents is overseen by the Lagos State Ministry of Health. - It is also true that the Federal Ministry of Health also play a role in the health of the residents. Please clarify what is meant by 'overseen'

Line 134 – 135: Government-owned health facilities include 18 PHC centres and the Lagos State University Teaching Hospital (LASUTH). - Is there no secondary health facility in the LGA? Mention the presence or absence of secondary health facility in the LGA.

Line 158 – 159: PHCs were selected based on having a high patient load and all eligible health workers in the selected facilities were recruited into the study - How many PHCs were selected in each LGA. The sampling method for selecting those PHCs not mentioned

Line 191: …. their first visit ….. - Does this mean their first ever visit/contact to facility or first visit for that episode of illness. Please clarify

Result – Analysis and Tables appropriate

Line 263 – 264: The mean ages at the baseline and the end of the study were 40.8±9.6 and 39.2±9.9 years respectively. - What is the explanation for the change in age at baseline and at the end of the study? If it is the 63 that completed the study that were used then their ages shouldn't have changed remarkably. Indicate the duration of each phase of the study in the methods section.

Line 270 – 271: …… level of knowledge was 270 high on symptoms, modes of transmission, and preventive measures - Provide the figure for the level of knowledge. It is not enough to state that it is high except in the discussion section.

Line 278 – 279: Marital status and LGA of health workers were statistically significantly associated with knowledge scores - In what direction are the differences?

Line 290: …….. interactions with trained PHC workers … - Provide information on the nature of their interactions (duration of interactions, mode of interaction - one - on - one or group health talk, etc) with PHC workers in the methods section.

Line 298 – 299: The interaction with trained PHC workers led to an increase in all aspects of the knowledge enquired about: - Provide data showing the increase as provided for the baseline

Well written in details with relevant tables and figures

Discussion – Well discussed with study limitations provided.

Line 326 - ….. cough. - delete - cough is repeated in this sentence

Conclusion – Clearly written with appropriate recommendation.

Study limitation - It is desirable to mention the limitation in the study design - The lack of control group

6. PLOS authors have the option to publish the peer review history of their article (what does this mean?). If published, this will include your full peer review and any attached files.

Reviewer #1: **Yes: **Nyemike Simeon Awunor

Reviewer #2: **Yes: **Prof. Tanimola Akande

---

## [Author Response · Author response to Decision Letter 0]

31 Aug 2022

A. Response to Journal requirements

Response. This has been done. The two files are labelled appropriately as well as the response to reviewers

2. PLOS ONE does not copy edit accepted manuscripts

Response. This has been done. The submission is free of typos and grammatical errors. 

3. Please include your full ethics statement in the ‘Methods’ section of your manuscript file. In your statement, please include the full name of the IRB or ethics committee who approved or waived your study, as well as whether or not you obtained informed written or verbal consent. If consent was waived for your study, please include this information in your statement as well

Response. Ethical approval was obtained from the Health Research Ethics Committee of the Lagos State University Teaching Hospital (REFERENCE NO: LREC/06/10/1501). Written informed consent was obtained from all the participants after explaining the study's aim and objectives. (This was in the initial submission, lines 258-260)

Upon re-submitting your revised manuscript, please upload your study’s minimal underlying data set as either Supporting Information files or to a stable, public repository and include the relevant URLs, DOIs, or accession numbers within your revised cover letter. For a list of acceptable repositories, please see http://journals.plos.org/plosone/s/data-availability#loc-recommended-repositories

Response. Data set is now uploaded as an Excel sheet

Response. Two new references have been added (no 16 and 21) making 24.

B. Responses to Reviewers (Line numbers in red are as on the tracked changes document)

1. The statistical analysis in general is appropriate and rigorous. However one is curious to know why an "independent t-test" was used to measure differences in knowledge, attitude and practice (KAP) pre- and post- intervention instead of a "paired t-test"?

Response. Line 269-272.The independent T test was used when we compared post training scores only for analysis of statistical association (Tables 4 and 5) but when we compared as on Table 3, the paired T test was used. This we believe is appropriate.

“Associations between the socio-demographic characteristics of respondents and their post-intervention COVID-19 knowledge, attitude, and preventive practices were assessed using an independent sample t-test and one-way analysis of variance (ANOVA) when data were normally distributed or Wilcoxon signed-rank and Kruskal-Wallis equality-of-populations rank test as appropriate when the data were not normally distributed” This was as in the first submission.

2. Line 89 – 90: Almost 80% had positive - had positive what?

82% used face masks - desirable to know what use of face mask means in the study. Is it consistent use of face mask or ever use of face mask?

Response. Line 86. Positive attitudes (80%) 

Line 87. Face masks use was consistent 

3. Line 127 – 128: The population is estimated to be over 20 million.- Provide reference

Response. Line 124. Reference 16 now included.

4. The health care of its residents is overseen by the Lagos State Ministry of Health. - It is also true that the Federal Ministry of Health also play a role in the health of the residents. Please clarify what is meant by 'overseen'

Response. The Statement has now been re-written

Line 125-127 “Health services for the citizens are available through both public and private facilities. The larger numbers of publicly owned facilities in Lagos State belong to the Lagos State Government although there are a few owned by the Federal Government”

5. Line 134 – 135: Government-owned health facilities include 18 PHC centres and the Lagos State University Teaching Hospital (LASUTH). - Is there no secondary health facility in the LGA? Mention the presence or absence of secondary health facility in the LGA.

Response. Line 135-136. There is no secondary health care facility in Ikeja LGA as the existing one was upgraded to become the teaching hospital (LASUTH).

6. Line 158 – 159: PHCs were selected based on having a high patient load and all eligible health workers in the selected facilities were recruited into the study - How many PHCs were selected in each LGA. The sampling method for selecting those PHCs not mentioned

Response. Line 163-165. PHCs were selected based on having a high patient load (four facilities in Ikeja LGA and five facilities in Alimosho LGA) and all eligible health workers in the selected facilities were recruited into the study. These PHC were selected by simple random sampling.

7. Line 191: …. their first visit ….. - Does this mean their first ever visit/contact to facility or first visit for that episode of illness. Please clarify

Response. Line 201 First visit for childcare during the study

8. Line 263 – 264: The mean ages at the baseline and the end of the study were 40.8±9.6 and 39.2±9.9 years respectively. - What is the explanation for the change in age at baseline and at the end of the study? If it is the 63 that completed the study that were used then their ages shouldn't have changed remarkably. Indicate the duration of each phase of the study in the methods section.

Response. Line 281. The observation by the reviewers was correct but we have now effected the appropriate changes. Only the 63 that completed had their menage computed. The baseline mean age initially used was for all the recruited participants, but this has now been removed see Table 1

Line 203,214,229-231.The duration of the phase of the study: baseline -four weeks; intervention four weeks. Post intervention data collection.one week. Now indicated in the methods section

9. Line 270 – 271: …… level of knowledge was 270 high on symptoms, modes of transmission, and preventive measures - Provide the figure for the level of knowledge. It is not enough to state that it is high except in the discussion section.

Response. Line 290-291.The details are now provided

The level of knowledge was high on symptoms (< 90%), modes of transmission (<80%), and preventive measures (<93%).

10. Line 278 – 279: Marital status and LGA of health workers were statistically significantly associated with knowledge scores - In what direction are the differences?

Response. Line 298-300. The direction has now been shown

Married PHC workers had higher mean knowledge scores compared with the single. PHC workers in Alimosho LGA had higher mean knowledge scores than those in Ikeja LGA.

11. Line 290: …….. interactions with trained PHC workers … - Provide information on the nature of their interactions (duration of interactions, mode of interaction - one - on - one or group health talk, etc) with PHC workers in the methods section.

Response. Line 230-233. This has been done in the methods section

After the training, the same set of clients were exposed to the trained PHC workers for one-on one counselling on invitation to the PHC. Each session lasted for about ten minutes.

12. Line 298 – 299: The interaction with trained PHC workers led to an increase in all aspects of the knowledge enquired about: - Provide data showing the increase as provided for the baseline.

Response. Line 321-329. The data was provided as before on Table 7

The interaction with trained PHC workers led to an increase in clients’ knowledge of symptoms (<80%), modes of transmission (<80%) and preventive measures (<90%).

Single clients had significantly higher mean knowledge scores than the married but clients in Ikeja LGA had significantly higher knowledge scores than those in Alimosho LGA

13. Line 326 - ….. cough. - delete - cough is repeated in this sentence

Response. Line 336. The repeated word “Cough” is now deleted.

14. Study limitation - It is desirable to mention the limitation in the study design - The lack of control group

Response. Line 439-440. Now done

The lack of a control group is another limitation of the study.

Professor Olumuyiwa Odusanya (signed)

Lead and Corresponding Author

---

## [Decision Letter · Decision Letter 1]

6 Sep 2022

Building capacity of Primary Health Care workers and Clients on COVID-19: Results from a web-based training.

PONE-D-22-16429R1

Dear Prof Odusanya,

We’re pleased to inform you that your manuscript has been judged scientifically suitable for publication and will be formally accepted for publication once it meets all outstanding technical requirements.

Kind regards,

Kehinde Kazeem Kanmodi, BDS

Academic Editor

PLOS ONE

Reviewers' comments:

Reviewer's Responses to Questions

**Comments to the Author**

1. If the authors have adequately addressed your comments raised in a previous round of review and you feel that this manuscript is now acceptable for publication, you may indicate that here to bypass the “Comments to the Author” section, enter your conflict of interest statement in the “Confidential to Editor” section, and submit your "Accept" recommendation.

Reviewer #2: All comments have been addressed

2. Is the manuscript technically sound, and do the data support the conclusions?

Reviewer #2: Yes

3. Has the statistical analysis been performed appropriately and rigorously? 

Reviewer #2: Yes

4. Have the authors made all data underlying the findings in their manuscript fully available?

Reviewer #2: Yes

5. Is the manuscript presented in an intelligible fashion and written in standard English?

Reviewer #2: Yes

6. Review Comments to the Author

Reviewer #2: Suggested correction in the original manuscript has been corrected satisfactorily by the authors. The corrections requested in the various sections particularly in the Methods and Results section have been done. The areas requiring clarification have been clarified.

7. PLOS authors have the option to publish the peer review history of their article (what does this mean?). If published, this will include your full peer review and any attached files.

Reviewer #2: **Yes: **Tanimola Makanjuola Akande

---

## [Editor Report · Acceptance letter]

27 Sep 2022

PONE-D-22-16429R1 

Building capacity of Primary Health Care workers and Clients on COVID-19: Results from a web-based training.

Dear Dr. Odusanya:

I'm pleased to inform you that your manuscript has been deemed suitable for publication in PLOS ONE. Congratulations! Your manuscript is now with our production department. 

Kind regards, 

on behalf of

Dr. Kehinde Kazeem Kanmodi 

Academic Editor

PLOS ONE